# Accuracy of Delirium Screening Tools in Older People with Cancer—A Systematic Review

**DOI:** 10.3390/cancers15102807

**Published:** 2023-05-17

**Authors:** Francisco Miguel Martínez-Arnau, Andrea Puchades-García, Pilar Pérez-Ros

**Affiliations:** 1Department of Physiotherapy, Universitat de València, Gascó Oliag 5, 46010 Valencia, Spain; francisco.m.martinez@uv.es; 2Frailty and Cognitive Impairment Research Group (FROG), Universitat de València, Menendez Pelayo s/n, 46010 Valencia, Spain; 3Department of Nursing, Faculty of Nursing and Podiatry, Universitat de València, Menendez Pelayo s/n, 46010 Valencia, Spain; puchadesgarciaandrea@gmail.com

**Keywords:** cancer, aged, delirium, assessment, accuracy, psychometric properties, incidence, prevalence

## Abstract

**Simple Summary:**

One of the main complications of cancer is delirium, especially in advanced stages. Our aim is to determine which delirium screening instrument is the most accurate in older people with cancer. A systematic review was performed on 13 different assessment tools, reporting an incidence of delirium ranging from 14.3% to 68.3%. The Delirium Observation Screening Scale (DOSS) showed the best metric properties, followed by the Nursing Delirium Screening Scale (NuDESC), Confusion Assessment Method (CAM) and Memorial Delirium Assessment Scale (MDAS). Screening tools for delirium are heterogeneous, and there is a need to analyze metric properties exclusively in the older population as knowing the tools with the greatest diagnostic accuracy will enable physicians and nurses to make the correct choice for early detection of delirium. In this way, the most appropriate measures could be implemented to avoid harmful consequences.

**Abstract:**

Background: The increase in life expectancy worldwide has led to a larger population of older people, which in turn entails a rising prevalence of cancer. One of the main complications of cancer is delirium, especially in advanced stages. Objective: To determine which delirium screening instrument is the most accurate in older people with cancer. Methods: A systematic review was designed. A literature search was performed in MEDLINE, EBSCO and SCOPUS; additional records were identified by handsearching. Selection criteria were studies involving people with cancer and a mean sample age of 60 years or older, assessing delirium, and reporting the metric properties of the assessment instrument. Studies with post-surgical patients and substance abuse delirium were excluded. Results: From 2001 to 2021, 14 eligible studies evaluated 13 different assessment tools, reporting an incidence of delirium ranging from 14.3% to 68.3%. The Delirium Observation Screening Scale (DOSS) showed the best metric properties, followed by the Nursing Delirium Screening Scale (NuDESC), Confusion Assessment Method (CAM) and Memorial Delirium Assessment Scale (MDAS). Only two studies were considered to be at low risk of bias using the QUADAS-2 Tool. No study exclusively examined this population group. Conclusions: Screening tools for delirium are heterogeneous for older people with cancer, and there is a need to analyze metric properties exclusively in the older population. Registered on PROSPERO ID: CRD42022303530.

## 1. Introduction

The increase in life expectancy worldwide has led to a larger population of older people, which in turn entails a rising prevalence of cancer [1]. Indeed, oncological diseases are mostly diagnosed in people of advanced age and are currently the main cause of death in people over 65 years old [2].

One of the main complications of cancer is delirium [3], especially in advanced stages. Oncological disease encompasses metabolic abnormalities, metastatic disease, vascular disorders, paraneoplastic and autoimmune syndromes, and the promotion of inflammatory mediators, all of which are considered risk factors for delirium. Neurotoxicity related to cytostatic treatments, pain management, and other symptomatic treatments also increases the risk of delirium [4,5]. In addition, older people with cancer are more vulnerable due to changes derived from the aging process and a higher burden of comorbidities, such as diabetes, stroke, vascular diseases, dementia, frailty, and malnutrition, among others [6]. In older people, delirium can result in loss of function, increased cognitive impairment, loss of quality of life, and increased morbidity and mortality. In addition, the longer the duration and severity, the worse the consequences [7].

The prevalence of delirium ranges from 18% to 33% in general medical or oncology wards and 42% to 58% in palliative care units. In older people with cancer, reported prevalence ranges from 22% to 57% [8]; however, data show that up to 84.2% of delirium cases remain undiagnosed [8,9]. Physicians diagnose delirium based on criteria in the Diagnostic and Statistical Manual of Mental Disorders (DSM-V) [10] or the International Classification of Diseases (ICD-10), but different screening tools have been validated [11] for use by other healthcare professionals, including nurses and psychologists [12]. Specifically analyzing the available assessment instruments in older people with cancer would enable a more precise estimate of the true prevalence of delirium in this population [8]. There are more than 30 validated tools for screening, diagnosis or assessment of the severity of delirium [11], including everything from brief screening tools for settings such as intensive care or the emergency department, where the assessment needs to be performed in 2–3 min [13], to other, more comprehensive instruments that take 20–30 min [14] for settings such as inpatient wards or long-term care. Most instruments require short training by experienced psychiatrists, doctors, or nurses [11]. Therefore, depending on the setting, type of patient, assessment time, or team, the validated tool with the highest diagnostic accuracy would be chosen.

In addition to instruments validated in adults, there are some specific instruments for the pediatric population, but there is no distinction in the validation of the scales in younger versus older adults [11].

There have been reviews of screening instruments for delirium in hospitalized older people [15], in adults with cancer [8], and in palliative care [16,17,18]. Nevertheless, to our knowledge, there are no published systematic reviews of delirium screening instruments in older adults with cancer. The primary aim of this review was to determine which instrument has the best metric properties for detecting delirium in older people with cancer, and the secondary aim was to evaluate the incidence and prevalence of delirium in this population.

## 2. Materials and Methods

This review followed the methods laid out in the Cochrane Handbook for Systematic Reviews of Diagnostic Test Accuracy. The study protocol is registered in PROSPERO (ID: CRD42022303530).

### 2.1. Literature Search

The research question was defined in the PICO format (population/intervention/comparison/outcomes), with the following search strategy:

(((((Delirium OR (Deliri*) OR (Confusion*) AND ((Diagnosis) OR (Comprehensive Geriatric Assessment) OR (Method) OR (Diagnostic) OR (diagnostical) OR (Diagnosis) OR (Assessment) OR (Assessing) OR (Scale) OR (Test) OR (testing) OR (Tests) OR (Screen) OR (screening) OR (Measurement) OR (Measurements) OR (validation) OR (tool) OR (instrument) OR (Delirium Detection)) AND ((Aged) OR (Frail Elderly) OR (Older) OR (Elder*) OR (Geriatric patient) OR (Geriatric*)) AND ((Cancer) OR (Oncol*)))

A literature search was conducted in the electronic databases MEDLINE, Scopus, and EBSCO, over the period from database inception to 20 December 2021. Each database was searched using the terms shown as a single search term or in combination using Medical Subjects Headings (MeSH) with the Boolean operators AND/OR.

To identify possible additional articles, reference lists of all relevant articles were manually cross-referenced. The search for unpublished studies included an electronic search of trial records: current controlled trials (www.controlled-trials.com, accessed on 20 December 2021), the National Institute of Clinical Health Databases (clinicaltrials.gov, accessed on 20 December 2021), Mednar, as well as a review of the grey literature and Google Search.

### 2.2. Inclusion and Exclusion Criteria

In order to answer our research questions, we applied the following criteria to select records for inclusion in the review: (1) full-text articles published in English or Spanish; (2) peer-reviewed original studies with experimental (randomized controlled trials), observational, and cross-sectional designs; (3) studies assessing delirium in older people (mean age ≥ 60 years) with oncological diseases; (4) inclusion of a description in the Methods section of the instrument used to detect delirium; and (5) reporting of the metric properties of the assessment tool(s). Studies in animals, case reports, qualitative studies, letters to the editor, abstracts from conferences, books and doctoral theses were excluded.

### 2.3. Data Collection and Analysis

A web-based system was used to manage the screening process and remove any duplicate citations. Thereafter, two members of the review team (PP-R and AP-G) independently screened the titles and abstracts against our selection criteria and retrieved all relevant full-text reports. Any discrepancies between review authors were resolved by consensus with a third member of the review team (FMM-A).

The same review authors Independently extracted the following data from each article: date of publication, study design, country, number and characteristics of the older participants (age and sex, type of cancer and setting), description of incidence or prevalence of delirium, assessment scale, gold standard, and validity and reliability of the instrument in the sample. In case of disagreement between the two reviewers regarding the manuscripts and the data extracted from them, the other author (FMM-A) acted as adjudicator.

### 2.4. Assessment of Risk of Bias in the Selected Studies

The included studies were assessed using the QUADAS-2 tool, developed by the Cochrane Review Group to evaluate the quality of diagnostic accuracy studies [19]. Studies were rated as being at low, unclear, or high risk of bias according to applicability concerns in the following four domains: (1) patient selection; (2) index test(s); (3) reference standard; and (4) flow and timing (“high” ratings indicate a high risk of bias and thus lower methodological quality). Studies were considered to be of acceptable methodological quality if they had an unclear risk of bias or applicability concern in one or two of the four domains. Studies with a high risk of bias or applicability concern in any of the domains were not considered to be of satisfactory methodological quality.

The judgments were made independently by two review authors (AP-G and FMM-A), and any discrepancies were resolved through consensus in consultation with a third review author (PP-R).

## 3. Results

### 3.1. Characteristics of Studies

The literature search yielded a total of 10,373 articles and 4 additional records through other sources. After the study selection process, 14 articles were included in the analysis. The full study selection process is presented in a PRISMA flow chart (Figure 1).

In all included studies, any patient aged 18 years or older was eligible; none of the articles used old age as an inclusion criterion, although the mean age of all samples was over 60 years, ranging from 60.9 years [20] to 76 years [21].

Half the studies were European [20,21,22,23,24,25,26], three took place in Asia, two in Canada, one in Australia and one in the USA. The studies included participants with any type of advanced cancer who were recruited in hospital oncology services or palliative care units. The incidence/prevalence of delirium in the included studies ranged from 14.3% [20] to 68.3% [27]. Fifty percent (*n* = 7) of the studies had a higher proportion of men than women, but some did not indicate data regarding sex (Table 1).

The metric properties of 13 assessment instruments were analyzed: the most commonly used was the Memorial Delirium Assessment Scale (MDAS) [27,28] and its versions in Spanish [21], Thai [29], Korean [30] and Italian [24]. Grassi also analyzed the Italian versions of the Delirium Rating Scale (DRS-98) and Mini Mental State Examination (MMSE).

Two studies assessed the Nursing Delirium Screening Scale (Nu–DESC) [28,31], and two the Delirium Observation Screening Scale (DOSS) [22,25].

The least commonly used instruments were the Confusion Assessment Method (CAM) [26,32], the Single Question in Delirium (SQiD) [32], and its Spanish translation [20]. The rest of the instruments were analyzed only in single studies (Table 2). The professionals in charge of using the assessment instruments were mainly nursing staff, although there are also studies where physicians, caregivers, psychologists, palliative and clinical staff were the evaluators (Table 2).

**Table 1 cancers-15-02807-t001:** Characteristics of included studies.

Study	Country	Type of Cancer	Setting	n	Mean Age (SD or Range)/% Men	Prevalence (%)
Barahona et al., 2018 [21]	Spain	All, advanced cancer	Hospice and general hospital	60	76 (69–83)/48	41.8
De la Cruz et al., 2015 [31]	USA	All, advanced cancer	Home, receiving hospice care	78	69 (49–91)/55	44, MDAS 42, Nu–DESC Nurse 24, Nu–DESC Caregiver evening 15, Nu–DESC Caregiver night
Detroyer et al., 2014 [22]	Belgium	All, palliative care	PCU and MOW	48	72 (67.25–78)/62.5	22.9
Gaudreau et al., 2005 [28]	Canada	All, palliative care	Hemato–oncology/Internal medicine hospital unit	59	61 (15–92)/NA	35.59
Grandahl et al., 2016 [23]	Danish	All, oncology inpatient	MOW	81	68.5 (7.8)/42	33
Grassi et al., 2001 [24]	Italy	All, derived from psychiatrist consultation	PCU and MOW	105	67.7 (13.2)/52.4	62.8
Hamamo et al., 2015 [33]	Japan	All, palliative care	PCU and MOW	2343	69.1 (12.8)	19.9
Kang et al., 2018 [30]	Korea	All, advanced cancer	PCU	123	66.92 (12.09)/42.28	23.52
Klankluang et al., 2019 [29]	Thailand	All	PCU	194	63.9 (13.3)/51.5	51 (8.1 hyperactive; 38.4 hypoactive; 53.5 mixed)
Lawlor et al., 2000 [27]	Canada	All, advanced cancer	PCU	104	64.4 (10)/NA	68.3
Nefjees et al., 2019 [25]	Netherlands	All, advanced cancer	MOW	187	64 (12)/66.3	50.26
Ryan et al., 2009 [26]	Ireland	All, advanced cancer	PCU	52	69.19 (36–93)/46.15	29.41
Sancho–Espinosa et al., 2018 [20]	Spain	Patients with solid tumors	MOW	42	60.9 (1.9)/71.4	14.3
Sands et al., 2021 [32]	Australia	All	MOW	73	68 (60.5–78)/42	38 (14.8 hyperactive; 59.3 hypoactive; 22.2 mixed)

MDAS: Memorial Delirium Assessment Scale; MOW: medical oncology ward; Nu–DESC: Nursing Delirium Screening Scale; PCU: palliative care unit.

The gold standard was mainly the DSM diagnostic criteria in its different versions and the CAM scale, as well as the MDAS and DSR-98, with the responsible professionals including physicians, nurses, psychiatrists or researchers (Table 2).

All instruments were assessed for sensitivity and specificity. Four studies did not report other measures, such as PPV and negative predictive value [23,26,27,28]. Validity, reliability, and other data such as area under the receiver operating curve (AUC) are also reported in Table 2. The instrument with the best diagnostic metric properties was the DOSS, assessed by Neefjes et al. [25] as having sensitivity, specificity and negative predictive values over 99% and PPV values of 95%. The instruments yielding the lowest sensitivity were the Nu-DESC scale, for use by caregivers [31], and the SQiD and CAM, for use by clinical staff [32].

Analyses of the Spanish, Thai, Korean and Italian versions of the MDAS found the highest sensitivity and specificity with different cutoffs: ≥7 points in studies by [21,27,28], ≥9 points in studies by Kang et al. and Klankluang et al. [29,30], and ≥13 points in the study by Grassi et al. [24] (Table 2).

### 3.2. Risk of Bias

After analyzing methodological quality with QUADAS-2, only two studies were deemed to be at low risk of bias on all domains and of low concern in terms of applicability [23,25]; one presented the instrument with the highest sensitivity and specificity [25].

Eight studies (57.14%) had excellent or acceptable methodological quality, while the other six (42.86%) had shortcomings due to a lack of generalizability to all older people with cancer. The sample selected by Grassi et al. [24] was previously screened via psychiatric consultation. Klankluang et al. [29] excluded patients with dementia, mental retardation, coma, or communication problems, while Ryan et al. [26] excluded patients who were terminally ill, unconscious or had communication problems, and repeated the assessment if participants had disorganized thinking or behavioral problems. Sancho-Espinosa et al. [20] included only solid tumors, not hematological disease, and the SQiD question was not validated in Spanish. Hamano et al. [33] analyzed only one item from a validated instrument, and Lawlor et al. [27] presented data from assessments in the same patients at different time points and carried out by different staff (Table 3 and Figure 2).

## 4. Discussion

In addition to being a neuropsychiatric syndrome, delirium can derive from the oncological pathophysiological process itself and is a possible side effect of the different cytostatic, radiotherapeutic and surgical treatments or interventions for symptom management. The prevalence of delirium increases in elderly oncological patients, in whom additional factors related to the aging process, such as advanced age and comorbidity, are also at play. Early detection of delirium enables prompt, appropriate treatment, and the literature suggests that the earlier treatment is administered, the less severe the consequences [34]. The most effective treatment is based on multi-component (pharmacological and non-pharmacological) interventions including re-orientation, appropriate lighting and noise levels, massages, and early mobility. This helps to reduce its duration and severity and, in turn, fatal consequences [34]. Early detection enables prompt treatment. This systematic review aimed to determine which delirium assessment tool has the best metric properties in older people with cancer, as well as to determine the prevalence of this syndrome in that population. The instrument with the highest sensitivity, specificity, positive predictive value, and negative predictive value was the DOSS.

### 4.1. Incidence

The incidence of delirium in the included studies ranged from 14% to 68%. Reviews estimate an incidence in older cancer patients of 22% to 57%, and more specifically in older patients in palliative care units, of up to 30% [8]. This wide range may be due to heterogeneous inclusion criteria or study settings. The lowest rate in our included studies was in older people with solid tumors in an inpatient oncology service [20], while the highest rates were after psychiatric consultation [24] and in patients with any type of advanced tumor [27]. The end-of-life process and the presence of psychiatric disease or disorders increase the risk even more and could explain the high incidence.

Previous studies such as Watt et al. [17] found higher rates in palliative care units, although the studies analyzed also include patients with diagnoses other than cancer. Van Velthuijsen et al. [15] also found higher rates in older people hospitalized in general medicine wards. The heterogeneity of included studies in terms of selection criteria, setting, screening instruments and comorbidity make any comparison difficult, although the higher the presence of risk factors, the higher the figures.

### 4.2. Assessment Tools

The 14 included studies evaluated 13 assessment tools. The DOSS showed the best predictive capacity in the study by Neefjes et al. [25], although Detroyer et al. [22] reported lower sensitivity, specificity, and positive and negative predictive values using the same cutoff (≥3 points) and the same assessor (nurses). The study sample or setting may interfere with this difference, and it may also be possible to analyze the gold standard used in each case.

The most widely used instrument was the MDAS, although it was not the one with the best metric properties. Despite being the most widely used in several languages, it is difficult to compare results, as the analyses used different cutoffs. Indeed, the reported predictive capacity varied widely, with the sensitivity of the MDAS ranging from 68% with a cutoff of ≥13 [24] to 97% with a cutoff of ≥7 [27]. Other studies used cutoffs ≥9 [29,30]. In validation of this scale, Breitbart et al. [35] proposed a cutoff of ≥13, citing a sensitivity of 70.59% and specificity of 93.75%.

The Nu-DESC is another widely used scale, validated in different languages in palliative care, yielding a sensitivity of 85.7% [15,36,37]. However, the use of different cutoffs modifies its sensitivity [31].

Despite being the most widely used instrument worldwide, only two studies analyzed the CAM scale [38]: Ryan et al. [26] obtained very good metric properties in palliative care units, while Sands [32] reported unacceptable sensitivity in an inpatient oncology service. Grassi et al. [24] analyzed two cutoffs in the Italian version of the DRS, with moderate results. This scale is primarily used for assessing delirium severity, with no explicit cutoff point for initial validation [39].

The tools with the lowest predictive capacity were the CDT, the Minicog and the DST, although combining them increased their sensitivity [23]. The SQID also showed unacceptable sensitivity rates in its validation study [40] and its (unvalidated) Spanish translation [20].

In addition to participants’ sociodemographic characteristics, the setting and the type or stage of cancer may also have influenced the results. The wide use of different instruments with different cutoff points, the construct itself, or the assessor (ranging from caregivers to clinical staff, nurses or physicians with or without previous education or training) are other factors that may affect accuracy. For example, de la Cruz et al. [31] obtained different incidence rates depending on the instrument used, the people in charge of carrying it out, and the time of the assessment, obtaining lower prevalence and lower sensitivity when caregivers performed the assessments instead of nurses, or when these took place at night instead of the evening.

Finally, few studies are at low risk of bias in older people with cancer. There is a need for studies that analyze the incidence and prevalence of delirium as well as all the factors that could influence its onset, to favor early identification and better outcomes.

### 4.3. Strengths and Limitations

This systematic review is the first to compare the diagnostic accuracy of delirium screening tools in older people with cancer. The use of the DSM or ICD for diagnosis is limited to physicians, while the use of validated assessment instruments allows the rest of the healthcare team to screen for delirium in this vulnerable population.

Previous studies have concluded that there is a need for rapid, useful tools for detecting delirium [9]. Given the existence of multiple delirium screening tools, understanding the diagnostic accuracy of each in the target population or setting of implementation allows for an informed choice.

The main limitation of the review evidence resides in the lack of studies exclusively in older people. In addition, the studies were heterogeneous in terms of population, setting, cancer types, and stages, precluding a robust comparison of the results. Furthermore, some studies analyzed modified tools or different cutoff points and used different gold standards or assessors, further complicating matters. In addition, we reviewed a limited number of databases and only studies published in English or Spanish were included.

### 4.4. Implications for Practice

The results of this systematic review raise several points to consider in future research into delirium in older people with oncological disease:Studies should include only older people with oncological disease in order to determine the specific predictive capacity of test(s) in this population and to analyze the results by age group.Comorbidities, hospital unit, and type of cancer should be analyzed in addition to possible risk factors derived from cytostatic treatment, radiotherapy, or other healthcare interventions.Only validated scales, such as the DOSS, CAM or Nu-DESC, should be used whenever possible, respecting the validated cutoff points.Whenever possible, the metric properties of the instrument should be measured in the population analyzed.Details of the assessment should be described, including the time when it is performed, the person carrying it out, and their previous training or experience in the use of the instrument.Studies should be designed to minimize the risk of bias in order to enable extrapolation of the data to the entire older population with oncological disease.

## 5. Conclusions

This is the first review to analyze screening instruments for delirium in older people with cancer. The tool with the best diagnostic accuracy is the DOSS, with a cutoff of ≥3 points. Prevalence of delirium ranges from 14% to 68% in settings including inpatient oncology services, palliative care units, and home care. Future studies should exclusively include older people and standardize the use of assessment tools.

## Figures and Tables

**Figure 1 cancers-15-02807-f001:**
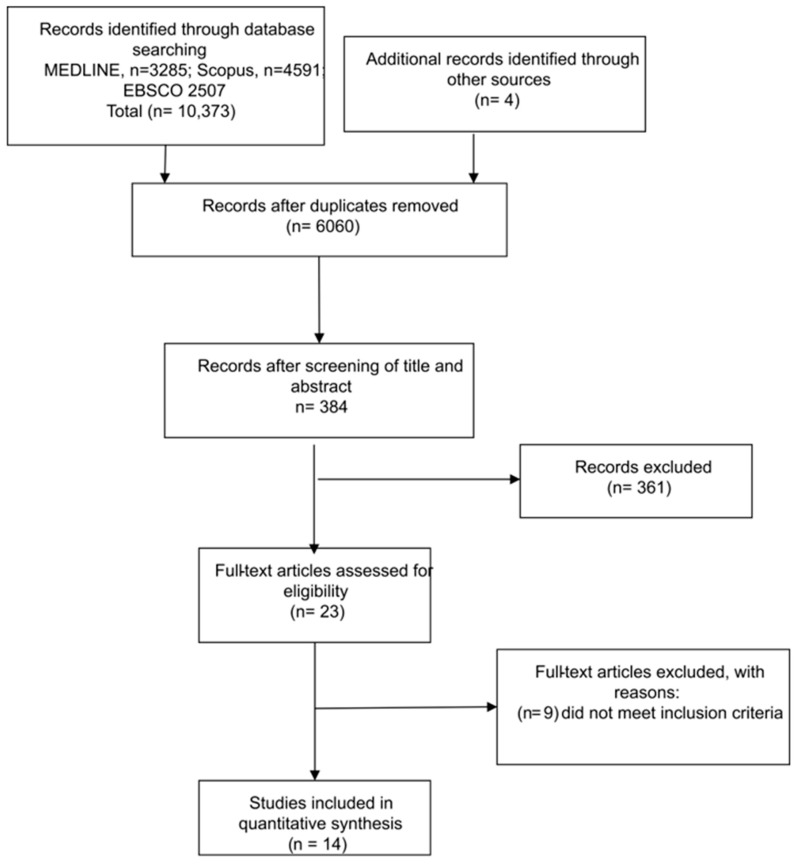
Preferred reporting items for systematic reviews and meta-analyses (PRISMA) workflow for literature search.

**Figure 2 cancers-15-02807-f002:**
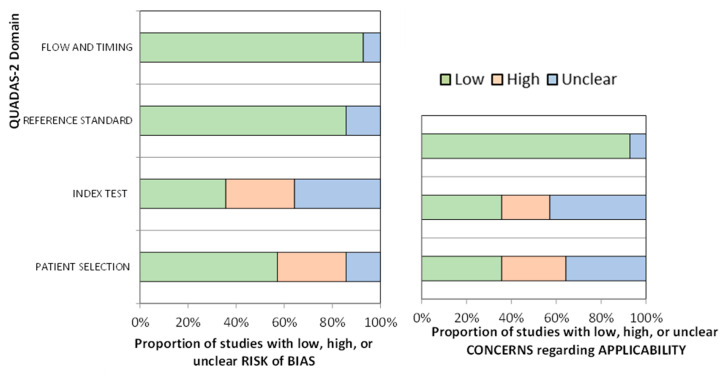
Risk of bias and applicability concerns of included studies, according to QUADAS-2.

**Table 2 cancers-15-02807-t002:** Screening instruments, gold standard and analyzed metric properties of included studies.

Study Characteristics		Validity	Reliability	Sensitivity % (95% CI)	Specificity % (95% CI)	PPV % (95% CI)	NPV % (95% CI)	AUC (95% CI)	Accuracy % (95% CI)
**Barahona et al., 2018** [21]									
Diagnostic instrument/cutoff	MDAS–S/ ≥ 7	NA	NA	92.9	71.8	70.2	93.3	0.93	NA
Assessor	Physicians								
Gold standard	CAM								
Assessor	Physicians								
**De la Cruz et al., 2015** [31]									
Diagnostic instrument/cutoff	Nu–DESC/ ≥ 2	NA	NA					NA	NA
Assessor	Nurse			63	67	61	68		
	Caregiver evening			35	80	58	61		
	Caregiver night			21	85	50	59		
Gold standard	MDAS								
Assessor	Trained nurse								
**Detroyer et al., 2014** [22]									
Diagnostic instrument/cutoff	DOSS/ ≥ 3	DOSS α = 0.772 DOSS and DI ρ = 0.53	NA	81.8 (52–95)	96.1 (90–98)	69.2 (42–87)	98 (93–99)	0.93 (0.82–1)	94.7 (89–98)
Assessor	Nurses								
Gold standard	CAM/DI								
Assessor	Nurses/researchers								
**Gaudreau et al., 2005** [28]			K = 0.89 (0.75–1) ^a^						
Diagnostic instrument/cutoffs	NuDESC/ > 1	NA		85.7 (65.4–95)	86.8 (72.7–94.3)	NA	NA	0.90	NA
	CRS/ > 0			76.2 (54.9–89.4)	81.6 (66.6–90.8)			0.83	
	CRS/ > 1			47.6 (28.3–67.6)	97.4 (86.5–99.5)				
	DSM–IV/ > 0			90.5 (71.1–97.4)	100 (90.8–100)			0.95	
	MDAS/ ≥ 7			95.2 (77.3–95.8)	89.5 (75.9–99.2)			0.97	
Assessor	Nurses								
Gold standard	CAM								
Assessor	Nurses and a Psychiatrist								
**Grandahl et al., 2016** [23]									
Diagnostic instrument/cutoff	CDT/ = 1 ^b^	NA	NA	81	46	NA	NA	NA	NA
	MiniCog/ < 3			67	85				
	DST/ ≤ 6			85	60				
	MiniCog and DST			93	60				
	CDT and DST			82	67				
Assessor	Nurses and physicians								
Gold standard	CAM								
Assessor	Psychiatrist								
**Grassi et al., 2001** [24]									
Diagnostic instrument/cutoff	DRS–I/ ≥ 10; ≥12	DRS α = 0.7	ρ = 0.76 ^c^, DRS & MDAS ρ = 0.88, MMSE & MDAS ρ = 0.67, MMSE & DRS	95; 81	61; 76	80; 85	89; 70	NA	NA
	MDAS–I/ ≥ 13	α = 0.89		68	94	95	63		
	MMSE < 24			96	38	88	72		
Assessor	Psychologist								
Gold standard	DSM–III								
Assessor	Neurologist or psychiatrist								
**Hamano et al., 2015** [33]									
Diagnostic instrument/cutoff	CCS 0/123	NA	NA	93.2 (90.6–95.1)	70.5 (69.9–71.0)	43.9 (42.7–44.8)	97.7 (96.8–98.3)	NA	75.0 (74.0–75.7)
	CCS 01/23			76.7 (73.4–79.7)	89.3 (88.5–90.0)	64.0 (61.–66.5)	93.9 (93.1–94.7)		86.8 (85.5–88.0)
Assessor	Physician								
Gold standard	DSM–IV								
Assessor	Physician								
**Kang et al., 2018** [30]									
Diagnostic instrument/cutoff	MDAS–K/ ≥ 9	α = 0.942	MDAS–K and DRS r = 0.95; ICC = 0.98	95.8	92.1	79.3	98.6	0.98 (0.96–1.00)	NA
Assessor	Palliative care staff								
Gold standard	CAM and DSM–IV								
Assessor	Psychiatrist								
**Klankluang et al., 2019** [29]									
Diagnostic instrument/cutoff	MDAS–T/ ≥ 9	α = 0.96 Content validity = 0.97 Content item validity from 0.67 to 1	ICC 0.98 (0.96–0.99)	92 (85–96)	90 (82–94)	90	91	0.91 (0.86–0.94)	Mild 42.6 Moderate 44.8 Severe 78.9 Profound 92.6
Assessor	Clinical psychologist								
Gold standard	CAM–ICU–T/DSM–V								
Assessor	Research assistant/psychiatrist								
**Lawlor et al., 2000** [27]									
Diagnostic instrument/cutoff	MDAS/ ≥ 7 ^d^	NA	ICC from 0.69 to 1	97	95	NA	NA	NA	NA
Assessor	Physician resident Medical staff Family interviewers								
Gold standard	DSM–IV								
Assessor	Physician resident								
**Nefjees et al., 2019** [25]									
Diagnostic instrument/cutoff	DOSS/ ≥ 3	NA	NA	>99.9 (95.8–100)	99.5 (95.5–99.9)	94.6 (88–97.7)	>99.9 (96.1–100)	NA	NA
Assessor	Bedside nurses								
Gold standard	DRS–R–98								
Assessor	Trained independent assessor								
**Ryan et al., 2009** [26]									
Diagnostic instrument/cutoff	CAM/(1 and 2) and (3 or 4)	NA	NA	96 (78–100)	93 (77–99)	NA	NA	NA	NA
Assessor	NCHDs								
Gold standard	DSM–IV								
Assessor	Psychiatrist								
**Sancho–Espinosa et al., 2018** [20]									
Diagnostic instrument/cutoff	Spanish SQiD/yes	Accordance of 95.2%; K = 0.88	NA	83.4 (43.6–97)	83.4 (68.1–92.1)	45.5 (21.3–72)	96.8 (83.8–99.4)	NA	83.4 (69.4–91.7)
Assessor	Nurse								
Gold standard	CAM and DSM–IV								
Assessor	Consultor team								
**Sands et al., 2021** [32]									
Diagnostic instrument/cutoff	SQiD/yes	K = 0.34 (0.01–0.56), SQID vs. DSM	NA	44.4 (25.2–64.7)	87 (73.7–95.1),	66.7 (45.9–82.5)	72.7(65.1–79.2)	NA	NA
	CAM	K = 0.32 (0.11–0.52), CAM vs. DSM		26.1 (10.2–48.4)	100 (92–100)	100 (100)	72.1 (67–76.7)		
Assessor	Clinical staff								
Gold standard	Psychiatrist interview								
Assessor									

^a^ Inter-reliability of CAM between nurses and researchers. ^b^ Cutoff points not specified in the study. Those provided in the referenced literature are given. ^c^ DRS: Intern-total correlation ranged from 0.09 (item 10) to 0.56 (item 7) and MDAS: Intern-total correlation ranged from 0.43 (item 7) to 0.82 (item 1) ^d^ The n is 104 but analyses include 330 assessments made at different points in time. It analyzes several cut-off points; those chosen by Lawlor et al. [27], in their conclusion are expressed in the table; CAM: Confusion Assessment Method; CAM–ICU: Confusion Assessment Method for the Intensive Care Unit; CCS: communication capacity scale; CDT: Clock Drawing Test; CRS: Confusion Rating Scale; DI: delirium index; DOSS: Delirium Observation Screening Scale; DRS–R–98: Delirium Rating Scale, Revised-98; DSM: Diagnostic and Statistical Manual of Mental Disorders; DST: Digit Span Test; MDAS: Memorial Delirium Assessment Scale; MDAS-I: Memorial Delirium Assessment Scale, Italian version; MDAS-K: Memorial Delirium Assessment Scale Korean version; MDAS-S: Memorial Delirium Assessment Scale Spanish version; MDAS-T: Memorial Delirium Assessment Scale Thai version; MiniCog: Mini Cognitive Test; MMSE: Mini Mental State Examination; NA: not available; NCHDs: non-consultant hospital doctors; NuDESC: Nursing Delirium Screening Scale; SQiD: Single Question in Delirium; α: Cronbach’s alpha; ρ: rho de spearman; K: Kappa Index; ICC: interclass correlation coefficient.

**Table 3 cancers-15-02807-t003:** Risk of bias and applicability concerns of included diagnostic test accuracy studies, according to the QUADAS-2 tool.

Study	Risk of Bias				Applicability Concerns			
	Patient Selection	Index Test	Reference Standard	Flow and Timing	Patient Selection	Index Test	Reference Standard	Overall Risk of Bias
Barahona et al., 2018 [21]	☺	☺	☺	☺	?	☺	☺	Acceptable
De la Cruz et al., 2015 [31]	☺	☺	☺	☺	?	☺	☺	Acceptable
Detroyer et al., 2014 [22]	☺	☺	☺	☺	?	☺	☺	Acceptable
Gaudreau et al., 2005 [28]	☺	?	☺	☺	☺	?	☺	Acceptable
Grandahl et al., 2016 [23]	☺	☺	☺	☺	☺	☺	☺	Excellent
Grassi et al., 2001 [24]	☹	?	☺	☺	☹	?	☺	Unsatisfactory
Humamo et al., 2015 [33]	☺	☹	☺	?	☺	?	☺	Unsatisfactory
Kang et al., 2018 [30]	?	☺	☺	☺	?	☺	☺	Acceptable
Klankluang et al., 2019 [29]	☹	?	☺	☺	☹	?	☺	Unsatisfactory
Lawlor et al., 2000 [27]	☺	☹	☹	☹	☺	?	?	Unsatisfactory
Nefjees et al., 2019 [25]	☺	☺	☺	☺	☺	☺	☺	Excellent
Ryan et al., 2009 [26]	☹	?	?	☺	☹	?	?	Unsatisfactory
Sancho-Espinosa et al., 2018 [20]	☹	?	☺	☺	☹	?	☺	Unsatisfactory
Sands et al., 2021 [32]	☺	?	☺	☺	☺	?	☺	Acceptable

☺ Low Risk; ☹ High Risk; ? Unclear Risk.

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
