# Peer review of "Accuracy of Delirium Screening Tools in Older People with Cancer—A Systematic Review"

_cancers, 2023, doi:10.3390/cancers15102807_

Round 1

Reviewer 1 Report

Dear authors,

congratulations to your work. Your topic is current and contributing to the your field. Prior the assessment I have checked plagiarism analysis and it is 11.5 which is great for literature review. 

Abstract - please do not use abbreviations in abstract. 

Introduction - I suggest you to better describe scientific contribution at the end of introduction

Methods and results well described.

Discussion and conclusion well written.

Literature is current.

Quality of English language is good. 

Author Response

Dear authors,

congratulations to your work. Your topic is current and contributing to the your field. Prior the assessment I have checked plagiarism analysis and it is 11.5 which is great for literature review. 

We thank the reviewer for the critical review and valuable comments, which we have taken into account in this revised manuscript. Itemized responses are listed below. All the modifications have been marked with track changes throughout the manuscript to facilitate review.

Abstract - please do not use abbreviations in abstract. 

Author’s answer: Thank you; we have now spelled out the abbreviations.

Introduction - I suggest you to better describe scientific contribution at the end of introduction

Author’s answer: Thank you; we have added the following:

“There are more than 30 validated tools for screening, diagnosis or assessment of the severity of delirium [11], including everything from brief screening tools for settings such as intensive care or the emergency department, where the assessment needs to done in 2–3 minutes [13] to other, more comprehensive instruments that take 20–30 minutes [14] for settings like inpatient wards or long-term care. Most instruments require short training by experienced psychiatrists, doctors, or nurses[11]. Therefore, depending on the setting, type of patient, assessment time, or team, the validated tool with the highest diagnostic accuracy would be chosen".

“In addition to the instruments validated in adults, there are some specific instruments for the pediatric population, but there is no distinction in the validation of the scales in younger versus older adults [11]”

Methods and results well described.

Discussion and conclusion well written.

Literature is current.

Author’s answer: Thank you.

Reviewer 2 Report

Thank you for giving me the opportunity to check the systematic review for delirium in cancer patients titled "Accuracy of delirium screening tools in older people with cancer. A systematic review". It's interesting for me to understand the status of delirium of cancer patients. 

We sometimes meet delirium in older patients with cancer in clinical. The score which can evaluate delirium is very important tool because delirium affects to their prognosis I think. But, in fact, physicians often do not know which score is best way to assess for delirium. In this systematic review, DOSS is the best way to assess for delirium from many kinds of previous reports about delirium from the data with the highest sensitivity, specificity, positive predictive value, and negative predictive value with cutoff of ≥ 3 points. This scoring has some limitations, but it's feasible way to assess delirium. 

Author Response

Thank you for giving me the opportunity to check the systematic review for delirium in cancer patients titled "Accuracy of delirium screening tools in older people with cancer. A systematic review". It's interesting for me to understand the status of delirium of cancer patients. 

We sometimes meet delirium in older patients with cancer in clinical. The score which can evaluate delirium is very important tool because delirium affects to their prognosis I think. But, in fact, physicians often do not know which score is best way to assess for delirium. In this systematic review, DOSS is the best way to assess for delirium from many kinds of previous reports about delirium from the data with the highest sensitivity, specificity, positive predictive value, and negative predictive value with cutoff of ≥ 3 points. This scoring has some limitations, but it's feasible way to assess delirium. 

Author’s answer: Thank you for your valuable comment.

Reviewer 3 Report

The article entitled "Accuracy of delirium screening tools in older people with cancer. A systematic review" is very interesting, well written and well constructed. I have only a few comments to make.

In the abstract, abbreviations such as DOSS, Nu-DESC ... should be defined.

In the introduction or the discussion, I suggest to quote scales that assess "pre-delirium" such as the Edmonton Symptom Assessement System and scales that assess "sickness behaviour".

Can we have an idea of how long it takes to complete the main tests and if special training is required beforehand?

In the discussion, I suggest mentioning the basics of therapeutic measures (improving light, massages, shortening hospitalization ...) to be applied once the diagnosis is established.

Author Response

The article entitled "Accuracy of delirium screening tools in older people with cancer. A systematic review" is very interesting, well written and well constructed. I have only a few comments to make.

We thank the reviewer for the critical review and valuable comments, which we have taken into account in this revised manuscript. Itemized responses are listed below. All the modifications have been marked with track changes throughout the manuscript to facilitate review.

In the abstract, abbreviations such as DOSS, Nu-DESC ... should be defined.

Author’s answer: Thank you for your comment; we have spelled out the abbreviations.

In the introduction or the discussion, I suggest to quote scales that assess "pre-delirium" such as the Edmonton Symptom Assessement System and scales that assess "sickness behaviour".

Can we have an idea of how long it takes to complete the main tests and if special training is required beforehand?

Author’s answer: Thank you for your comment. We have added the following to the Introduction:

“There are more than 30 validated tools for screening, diagnosis or assessment of the severity of delirium [11], including everything from brief screening tools for settings such as intensive care or the emergency department, where the assessment needs to done in 2–3 minutes [13] to other, more comprehensive instruments that take 20–30 minutes [14] for settings like inpatient wards or long-term care. Most instruments require short training by experienced psychiatrists, doctors, or nurses[11]. Therefore, depending on the setting, type of patient, assessment time, or team, the validated tool with the highest diagnostic accuracy would be chosen".

“In addition to the instruments validated in adults, there are some specific instruments for the pediatric population, but there is no distinction in the validation of the scales in younger versus older adults [11]”

In the discussion, I suggest mentioning the basics of therapeutic measures (improving light, massages, shortening hospitalization ...) to be applied once the diagnosis is established.

Author’s answer: Thank you for your comment. We have added the following to the Discussion:

“Early detection of delirium enables prompt, appropriate treatment, and the literature suggests that the earlier treatment is administered, the less severe the consequences. The most effective treatment is based on multi-component (pharmacological and non-pharmacological) interventions including re-orientation, appropriate lighting and noise levels, massages, and early mobility.”